# The Shortest Duration Constrained Hidden Markov Model: Data denoise and forecast optimization on the country-product matrix for the Fitness-Complexity Algorithm

Pengcheng Song[1], Xiangyu Zong[2]*, Ximing Chen[3], Qin Zhao[4], Lubingzhi Guo[5]

**1** School of Economics and Finance of Xi'an Jiaotong University, Xi'an, China, **2** University of Glasgow Business School, University of Glasgow, Glasgow, United Kingdom, **3** Chinese Academy of International Trade and Economic Cooperation, Beijing, China, **4** School of Mathematical Sciences, Peking University, Beijing, China, **5** School of Computing Science, University of Glasgow, Glasgow, United Kingdom

* x.zong.1@research.gla.ac.uk

**Data Availability Statement:** The export data is from COMTRADE database. https://comtrade.un.org/. This web provides the API to download the

## Abstract

The Economic Fitness Index describes industrial completeness and comprehensively reflects product diversification with competitiveness and product complexity in production globalization. The Fitness-Complexity Algorithm offers a scientific approach to predicting GDP and obtains fruitful results. As a recursion algorithm, the non-linear iteration processes give novel insights into product complexity and country fitness without noise data. However, the Country-Product Matrix and Revealed Comparative Advantage data have abnormal noises which contradict the relative stability of product diversity and the transformation of global production. The data noise entering the iteration algorithm, combined with positively related Fitness and Complexity, will be amplified in each recursion step. We introduce the Shortest Duration Constrained Hidden Markov Model (SDC-HMM) to denoise the Country-Product Matrix for the first time. After the country-product matrix test, the country case test, the noise estimation test and the panel regression test of national economic fitness indicators to predict GDP growth, we show that the SDC-HMM could reduce abnormal noise by about 25% and identify change points. This article provides intra-sample predictions that theoretically confirm that the SDC-HMM can improve the effectiveness of economic fitness indicators in interpreting economic growth.

## Introduction

Economic data are fundamental to economic research. Numeric value determines the accuracy of economic analysis in two key ways: first, the more uncomplicated the data composition required for economic research, the more essential the quality of economic data becomes; second, the more unsophisticated the value categories of economic data, the more crucial data denoise is for data quality. This work selects economic fitness and complexity analysis whose sole data input is the country-product matrix, and the matrix is binary data for which the value

data. Besides, All relevant data are within the paper and its Supporting Information files.

**Funding:** The authors received no specific funding for this work.

**Competing interests:** The authors have declared that no competing interests exist.

composed is extremely simple, only 0 and 1. The sole data required for economic fitness is measured by country-product matrix through the Revealed Comparative Advantage (RCA), whose original data are the countries' export data of specific products.

Economic Fitness (EF) measures both the industrial diversification of a country from the perspective of quantity, and the competitiveness of its products from the perspective of quality, which is reflected in where the product is in the Global Value Chains (GVCs). Gereffi [1] initially carried out research on global value chains based on the East Asian garment industry, which now refers to the processes of globally distributing the entire production. Products in the relatively high-end segment will have stronger competitiveness, meaning the country can obtain higher added value [2]. The status of a country's products in the production chains is relatively stable, and its evolution process also requires a certain period to perform. Excessive fluctuations of the RCA and country-product matrix are contrary to the objective laws of production globalization and cannot reflect changes in the essential competitiveness of products and countries.

The Economic Fitness method is developed by Tacchella et al. (2013) [3] for quantitative economic complexity analysis. It is used to measure the degree of product diversification with competitiveness and product complexity. Higher economic fitness indicates a more varied product structure, and shows that the country's exports are more risk-resistant. The Fitness-Complexity Algorithm is a non-linear iteration method (see Eq 3) which can produce different results for different countries, despite them having the same initial value. The RCA value and the Country-Product Matrix after noise reduction through the SDC-HMM support more stable and more objective recursive equations (Eq 3). The shortest duration constraint requires the hidden state time series to maintain the former state for at least a default number. In this paper, the shortest duration constraint t = 2 will help the model more in line with the actual economic situation. On the premise of retaining the original data as much as possible, the impact of data noise can be better reduced. The importance of the SDC-HMM performs in two aspects. First, setting a minimum duration longer than one period is conducive to making full use of data time dimension information and effectively removing abnormal data value noise. Second, setting the value of d to be greater than or equal to 2 is to fully retain the message of the data time dimension and identify the change of data trend in time. The denoise process can enable the pursuit of lower noisy product complexity, and then reduce noise in the recursive cycle. Therefore, the consequential noise of economic fitness is lower and a more accurate prediction of GDP growth can be obtained.

The latest research also finds that the economic complexity of a country is the reason for its long-term economic growth [4]. Ricardo's traditional comparative advantage theory [5] holds that a country should merely export a few products that adhere to its own comparative advantage and fulfil international specialization according to its comparative advantage. Nevertheless, the national export figures suggest that internationally accredited developed countries' export structure has remained fully diversified. The internal economic fitness of a country constituted by the diversity and competitiveness of its products corresponds to the hidden state of the model, while changes in the external export data of countries only correspond to fluctuations in the apparent state. It is indeed appropriate to apply the Hidden Markov Model to process country-product matrix data [6].

However, when the export data fluctuate sharply over a short period of time due to factors such as tariffs, exchange rates and other countries' trade policies, it does not necessarily result in a drastic change in the internal capacity of the country's economic fitness. Furthermore, the economic fitness index at time t is used to predict GDP at (t+1) and other future times. The export RCA data and the unstable state of the country-product matrix will not be able to meet the objective laws of relative stability and smoothness in global production, which in turn will

reduce the predictive effect of a country's economic fitness on its economic growth (GDP). In view of this, this paper imposes a shortest duration constraint on the Hidden Markov Model to achieve the corresponding requirements.

The main innovations of this paper are as follows: First, this paper innovatively uses the shortest duration constraint to achieve noise reduction and removal of ultra-short-term abnormal economic fluctuations, while ensuring the identification and confirmation of medium- and long-term economic trends. In the end, the time dimension information of the time series data can be fully utilized to effectively remove abnormal data noise; and the data time dimension information can also be fully retained, and the data trend change point can be identified in time. Without the shortest duration constraint, macroeconomic data, especially the higher frequency monthly data, will show greater volatility. Most of the fluctuations are caused by the disturbance of certain unexpected events, which cannot characterize the intrinsic economic trends.

Secondly, this article innovatively chooses a Country-Product Matrix of binary data and the Economic Fitness indicators to maximize the extent to which the SDC-HMM for macroeconomic time series data with noise reduction optimization will ultimately improve the results of economic analysis. As a macroeconomic analysis model, the country Economic Fitness analysis requires only one data indicator, i.e. country-product matrix, and the Country-Product Matrix is binary data with a single data composition, which results in any noise and fluctuations in the data being directly transmitted to the macroeconomic analysis results. Therefore, the object of this study is chosen to maximize the effect of data noise reduction and optimization of the SDC-HMM.

The possible contributions of this paper are mainly reflected in the following four aspects: First, the shortest duration constraint is added to the analysis of the hidden Markov model, making full use of the time dimension information to address the problem of changing points in the time series data, and providing a practical way. Second, with the national economic fitness analysis, in which two characteristics highlight: data required is single, and the data's value is extremely simple (binary data), we build a natural experiment to verify the data denoising effect of the SDC-HMM; Third, through the shortest duration constraint optimization, the prediction accuracy of economic fitness is improved. Fourth, it enriches the application of hidden Markov models in the field of economics and further expands the role of hidden Markov models in data cleaning.

## The Hidden Markov Model

The Hidden Markov Model (HMM) includes two sets of time series random variables. One group is a hidden state random variable time series, in which the state of the sequence cannot be observed; another group is an explicit state random variable time series, which is observable and the sequence is generated via a hidden Markov chain transmission. According to the above model, the Hidden Markov Model contains two sets of state time series and three sets of probability time series. The detailed composition sequence is described as follows [7]:

1. hidden state time series set H

   $H = \{h_1, h_2, \ldots, h_N\}$, which involves N states. The hidden state in time t can be any one in the hidden state set, denoted as $q_t$, and satisfies the condition $q_t \in \{h_1, h_2, \ldots, h_N\}$.

2. explicit state time series set X

   $X = \{x_1, x_2, \ldots, x_M\}$, which involves M states. The explicit state in time t can be any one in the explicit state set, denoted as $p_t$, and satisfies the condition $p_t \in \{x_1, x_2, \ldots, x_M\}$.

3. hidden state transition probability distribution A

   $A = (a_{ij})$, and $a_{ij} = P(q_{t+1} = h_j | q_t = h_i)$, where $1 \leq i \leq N$ and $1 \leq j \leq N$. $a_{ij}$ is the conditional probability of the state transition to the state $h_j$ at time (t+1) when the hidden state at t is $h_i$ (the transition probability of time t).

4. initial state probability distribution $\pi$

   $\pi = \{\pi_1, \pi_2, \ldots, \pi_N\}$, and $\pi_i = P(q_i = h_i)$, $1 \leq i \leq N$, the probability that the initial state of the hidden state time series becomes $h_i$ is $\pi_i$.

5. hidden state emission probability distribution E

   $E = \{e_i(x_l)\}$, and $e_i(x_l) = P(p_t = x_l | q_t = h_i)$, where $1 \leq i \leq N$, $1 \leq l \leq M$. $e_i(x_l)$ is the conditional probability of the state emission to the explicit state $x_l$ when the hidden state at t is $h_i$ (the emission probability of time t).

## The Shortest Duration Constrained Hidden Markov Model

1. the shortest duration constraint

   The shortest duration constraint requires the hidden state time series to maintain the former state for at least a default number $dz$,

$$d \geq t \tag{1}$$

   t is the shortest duration. Suppose that the hidden state is $q_t$ at time t, and the time sequence changes value at time t. The hidden state time series must meet the following equation

$$q_{(t-1)} \neq q_t = q_{(t+1)} = \ldots = q_{(t+z-1)} \tag{2}$$

2. the constrained Hidden Markov Model

   1) the constrained hidden state time series set $H'$

      $H' = \{h_{11}, h_{12}, \ldots, h_{1z}, \ldots, h_{Nz}\}$, which involves $(z \times N)$ states. The hidden state in time t can be any one in the hidden state set, denoted as $q_t$, and satisfies the condition $q_t \in \{h_{11}, h_{12}, \ldots, h_{Nz}\}$.

   2) the constrained explicit state time series set $X'$

      $X' = \{x_1, x_2, \ldots, x_M\}$, which involves M states. The explicit state in time t can be any one in the explicit state set, denoted as $p_t$, and satisfies the condition $p_t \in \{x_1, x_2, \ldots, x_M\}$. Compared with the HMM, the constrained HMM has the same explicit state time series set while the hidden state time series set is extended from the original N states to $(z \times N)$ states, where the set $\{h_{11}, h_{12}, \ldots, h_{1z}\}$ is consistent with $h_1$.

   3) constrained hidden state transition probability distribution $A'$

      $A' = (a_{ij,kl})$, and $a_{ij,kl} = P(q_{t+1} = h_{kl} | q_t = h_{ij})$, where $1 \leq i \leq N$, $1 \leq k \leq N$, and $1 \leq j \leq z$, $1 \leq l \leq z$. $a_{ij}$ is the conditional probability of the state transition to the state $h_{kl}$ at time (t+1) when the hidden state at t is $h_{ij}$ (the transition probability of time t).

   4) constrained initial state probability distribution $\pi'$

      $\pi' = \{\pi_{11}, \pi_{12}, \ldots, \pi_{1z}, \ldots, \pi_{Nz}\}$, and the same as HMM, $\pi_{ik}$ is the probability when initial state is $h_i$. $\pi_i = P(q_i = h_i)$, $1 \leq i \leq N$.

   5) constrained hidden state emission probability distribution $E'$

      $E' = \{e_{ij}(x_l)\}$, and $e_{ij}(x_l) = P(p_t = x_l | q_t = h_{ij})$, where $1 \leq i \leq N$, $1 \leq l \leq M$, $1 \leq j \leq z$. $e_{ij}(x_l)$

is the conditional probability of the state emission to the explicit state $x_l$ when the hidden state at t is $h_{ij}$ (the emission probability of time t).

It is important to carry out change point detection and subsequent noise reduction optimization of economic time series data. Data change point noise reduction was initially proposed by Page [8] in quality control research on continuous sampling inspection, and subsequently developed into a range of fields such as economics, big data, biology and finance. Change point detection refers to the analysis of historical data sequences to detect the presence of meaningless data change points such as abnormal values. To achieve data noise reduction and forecast optimization, time series models contain the Auto-Regressive and Moving Average Model (ARMA) and the Hidden Markov Model [7]. As a kind of time domain analysis method [9], Luong [10] proposed applying a part of conditional restrictions to the hidden state sequence through the Hidden Markov Model (HMM) method. The HMM was established in 1957 as a time series analysis model. The specific application was first proposed by Rabiner & Juang in 1986 and used to resolve problems in the field of language recognition. In this application, the HMM is employed to address the noise influence in the acoustic model. Subsequently, the HMM was widely used in the fields of natural sciences and engineering for biological sequence alignment, image processing and facial recognition [11–13]. In recent years, it has been gradually applied to the humanities and social sciences for economic forecasting, analysing online public opinion, and financial transactions. In finance and economics, Rossi & Gallo [14] employed the HMM to anatomize the volatility of financial asset returns and Huang Xiaobin et al. [15] used it to model and analyze the unpredictable stock information state. Tacchella et al. [6] apply the HMM to optimize noise reduction of country-product matrix data, performing preliminary processing by setting a certain probability that each RCA quartile value will be generated by each stage of development, but this method can still not effectively eliminate data noise. The development stage of a year with extremely high values in a certain year within the low-value period confirmed by the HMM will display a considerable deviation from virtual national economic fitness.

Aiming at data sanitation, this paper attempts to realize noise reduction and forecast optimization of country-product matrix data by imposing a shortest duration constraint on the Hidden Markov Model. The specific constraint process ensures the duration of the original state by setting the HMM hidden state time series number to be greater than or equal to a certain minimum value, so as to ensure the consistency of the time series data nodes and the original current, while the trend shift needs a certain duration to be confirmed.

## The Fitness-Complexity Algorithm

The Fitness-Complexity Algorithm is an iteration process, which includes two principal indicators of Economic Fitness and Product Complexity. The calculation is defined as follows.

$$
\begin{cases} \tilde{F}_c^{(n)} = \sum_p M_{cp} Q_p^{(n-1)} \\ \tilde{Q}_p^{(n)} = \dfrac{1}{\sum_c M_{cp} \dfrac{1}{F_c^{(n-1)}}}, \end{cases} \qquad \begin{cases} F_c^{(n)} = \dfrac{\tilde{F}_c^{(n)}}{\langle \tilde{F}_c^{(n)} \rangle_c} \\ Q_p^{(n)} = \dfrac{\tilde{Q}_p^{(n)}}{\langle \tilde{Q}_p^{(n)} \rangle_p} \end{cases} \tag{3}
$$

The iteration process is divided into two major steps. First, the Fitness of a country is weighted by the Complexity of product and the Complexity is inversely proportionate to the number of countries which can export the same product. Second, the intermediate variables $\tilde{F}_c^{(n)}$ and $\tilde{Q}_p^{(n)}$ are normalized into $F_c^{(n)}$ and $Q_p^{(n)}$ by the summed mean denominator. The initial condition value

is $\tilde{Q}_p^{(0)} = 1$ for any product and $\tilde{F}_c^{(0)} = 1$ for every country. The $M_{cp}$ is the binary Country-Product Matrix which is derived from the Revealed Competitive Advantage (see Eqs 4 and 5).

As shown in the recursive iteration, the Complexity of product in step $n$ is inversely proportionate to the reciprocal of the Fitness of a country in step $(n-1)$. As Economic Fitness rises, product complexity also rises in the iteration process. In addition, Economic Fitness in step $n$ is also directly proportionate to the Product Complexity of the previous step. In this way, the iteration serves as an anchor aiming at the explored economic potentiality and as an amplifier when the input of RCA or Country-Product Matrix involves noise data.

Revealed comparative advantage (RCA) is an indicator used to measure a country's product competitiveness and national comparative advantage [3, 16, 17]. The Heckscher-Olin model not only has corresponding difficulties when dealing with three-element models, such as labor, capital and raw material inputs, but the conclusions drawn by distinguishing the endowment when the number of countries is greater than 2 are inconsistent with facts [18]. To overcome the above-mentioned shortcomings, revealed comparative advantage (RCA) measures the competitiveness of a product through the export data of a specific product in a country, which is expressed as follows.

$$RCA_{c,p} = (\frac{E_{cp}}{\sum_p E_{cp}})/(\frac{\sum_c E_{cp}}{\sum_{c,p} E_{cp}}) \tag{4}$$

Where $RCA_{c,p}$ is the revealed advantage of country C in product P. $E_{cp}$ is the export of country C in product P within a certain time period. $\sum_p E_{cp}$ is the aggregate export of country C during the period. $\sum_c E_{cp}$ is the aggregate export of product from all the countries around the world. $\sum_{c,p} E_{cp}$ is the total world export. When the RCA value is greater than or equal to 1, then the product P of country C has a revealed comparative advantage, otherwise it does not. By binarizing the numerical value of revealed comparative advantage, we can obtain the country-product matrix, which is the $M_{cp}$ matrix. The relevant definitions are as follows.

$$M_{cp} = \begin{cases} 1 & if\ RCA_{c,p} \geq 1, \\ 0 & if\ RCA_{c,p} < 1 \end{cases} \tag{5}$$

The $M_{cp}$ matrix is a two-valued matrix, whose value is 1 when the RCA value is greater or equal to 1 and converts to 0 when the RCA value is below 1.

A country's economic competitiveness and development level are by and large positively related to its economic diversification, that is, economic complexity. Diversification is the dominant factor in the globalized economic market [6]. In studying how to quantify the level of a country's economic complexity, Hidalgo and Hausmann [2] first proposed the "Method of Reflections". With the binarization of the country-product matrix of export data, they apply iterative methods in establishing a linear relationship between the competitiveness of a country and product complexity to quantify the level of economic complexity. In order to overcome the related conceptual and mathematical defects of the linear relationship, Tacchella et al. [1] proposed the "Fitness-Complexity Algorithm" to define a country by the binarized country-product matrix. The nonlinear relationship between economic fitness and product complexity further optimizes the quantitative analysis of economic complexity.

It is particularly crucial to effectively reduce noise for low-fitness countries. They have fewer bits that are not "0" in the country-product matrix, so as one or more noisy pieces of data appear, it will be particularly difficult to correctly judge their fitness [6]. Furthermore, there is still a considerable degree of noise in the data obtained through the binarization

process, and especially when the RCA value fluctuates around the "1" threshold, the accuracy of the country-product matrix is difficult to guarantee.

The authenticity change of the internal trend also needs to be confirmed by the minimum duration constraint. This restraint will not recognize the data change point whose duration is shorter than the minimum pre-setting of the constraint, with the data smoothed and restored through HMM, and finally obtains high-quality country-product matrix time series data by decoding the hidden state time series to optimize the quantitative analysis of complexity. Such constraints can also be found in Zhuang Yu and He Zhenfeng [19], in which the constrained HMM is applied for change point detection, and economic cycle prediction is performed by anatomizing simulation data and GNP data.

## Results

### Model training and decoding algorithm

In this paper, the country-product matrix constructed by RCA binarization is utilized as the explicit state. The explicit state time series is divided into low and high states, which correspond to the binary values "0" and "1" respectively. The data for each product in each country is trained separately with a SDC-HMM.

In this paper, the shortest duration constraint t = 2 is set to bring the model more in line with the actual economic situation. On the premise of retaining the original data as much as possible, the impact of data noise is further reduced. The economic meaning of this time restriction setting is reflected in the confirmation of the economic cycle, that is, in the macro-economic cycle. The negative growth state needs to last at least two time periods before economic recession can be identified [20].

The value of d chosen in this paper is assigned to be greater than or equal to 2, which has strong economic characteristics and implications. The choice of this value aims to improve the economic forecasting effect through two aspects. First, setting a minimum duration greater than one period is conducive to making full use of data time dimension information and effectively removing abnormal data value noise. As mentioned above, in the case of the export data used for the Economic Fitness analysis in this paper, for example, export data will fluctuate drastically in a short period due to factors such as tariffs, exchange rates, and trade policies of other countries, but such changes in data that fluctuate from period to period do not necessarily lead to drastic changes in the intrinsic capacity of the Economic Fitness. Economic analysis aims to characterize intrinsic economic development and predict future trends through data. The unstable state of export RCA data and Country-Product Matrix will not satisfy the objective law of relative stability and smoothing of global value chains, which in turn reduces the predictive effect of a country's Economic Fitness on its economic growth. By choosing the value of d to be greater than or equal to 2, the noise reduction of such anomalous noise data will be achieved.

Second, setting the value of d to be greater than or equal to 2 is to fully retain the information of data time dimension and identify the change of data trend in time. If the minimum duration constraint is set too long, the data variation points inherent in the economic data cannot be identified in time. If a larger value is set, the data variables that characterize the macroeconomic trend changes will be identified with a longer time lag. This identification time lag will affect the timeliness and accuracy of macroeconomic forecasting to a certain extent. The export data selected for the economic adaptation analysis in this paper is a kind of data that changes relatively frequently, and the relative values of factor endowments and comparative advantages of different countries change more rapidly. Given this, we choose to set the value of d to be greater than or equal to 2, to maximize the retention of data time dimension information and timely identification of economic change trends. The final research results also show the correctness of this setting.

Furthermore, as the leading indicators used by countries to predict economic trends, the Purchasing Managers 'Index(PMI), known as the"barometer"of industry and even the macro economy, can also prove that the economic significance of the shortest duration constraint. PMI conducts monthly surveys on purchasing managers of various departments through a questionnaire and collects, summarizes the comprehensive index. Taking China's PMI as an example, the 50% value is the economic "prosperity and dryness line". Provided that the PMI value of the month is lower than 50%, it means that economic expectations are sluggish, and if is higher than 50%, it indicates that the future economic development is in the expansion stage. The value and related economic significance are mostly comparable to the economic meaning of the country-product matrix in this article about the competitiveness of countries and products. For China's PMI, when the index changes in the same direction for three consecutive months, it can reflect the trend switch of the country's macro economy. More precisely, if the cycle is in a low-value period with only the single-period data rises above the theoretical line, it cannot be judged that the economy has fully recovered. To achieve a trend improvement in the economy, the same direction value that lasts for 3 months is needed to confirm.

The training algorithm is the Baum-Welch algorithm and the decoding algorithm is the Viterbi algorithm. The specific steps are given in Fig 1.

Fig 2 is the schematic diagram of the HMM and constrained HMM. First, the transition state time series data are obtained from the RCA raw data. Subsequently, the time series data of the development stage are obtained based on the calculation of the emission probability distribution. In this process, the constrained HMM requires that each state data node be consistent with its previous (or next) state data node to meet the minimum duration constraint t = 2.

## Test 1: Country-product matrix test

As shown in Fig 3, data analysis is performed by comparing the RCA raw data (RCA) and RCA binary noise reduction data (RCA Binarization). The dotted line is the original RCA data, and the solid line is the RCA binary noise reduction data. As can be clearly seen, when the original RCA data fluctuate at the "1" threshold, there is a lot of noise in the RCA binarized data. In Fig 4, the dotted line represents the original RCA data, and the solid line the constrained HMM Binarization data. Noise reduction analysis is performed by comparing the RCA original data and the constrained HMM binarized noise reduction data.

Comparative analysis of the two graphs indicates that the Hidden Markov Model with the shortest duration constraint can greatly reduce the fluctuation noise of the RCA raw data at

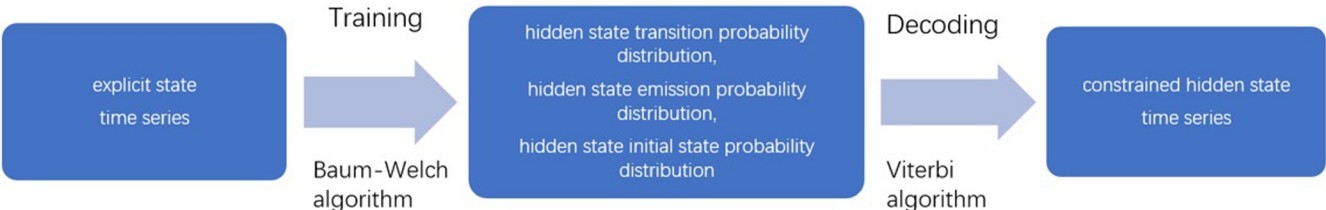

**Fig 1. The model training and decoding process of the SDC-HMM.** To demonstrate the steps of training and decoding we use the Baum-Welch algorithm and Viterbi algorithm respectively. Firstly, the explicit state time series are trained into the hidden state transition probability distribution, the hidden state emission probability distribution, and the hidden state initial state probability distribution. Then we perform the Viterbi algorithm in decoding these probability distributions into the constrained hidden time series.

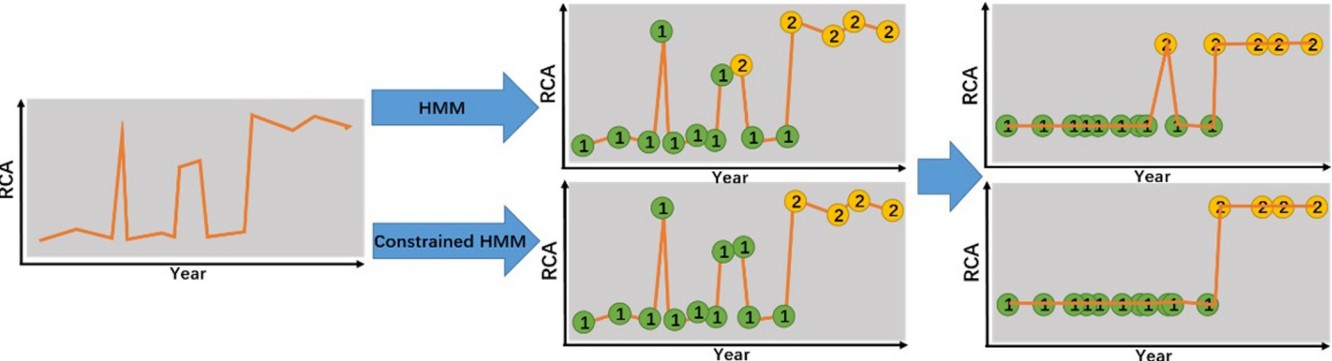

**Fig 2. The schematic diagram of the HMM and SDC-HMM.** The transition state time series data are obtained from the RCA raw data. Subsequently, the time series data of the development stage are obtained based on the calculation of the emission probability distribution.

the "1" threshold. Moreover, the constrained HMM can still effectively identify changes in the RCA's inherent trends, because the changing points can be effectively identified.

### Test 2: Country case test

In the noise reduction test of the constrained Hidden Markov Model in specific country cases, we find that the SDC-HMM not only presents the significant advantage of removing the noise of outlier data, but can also effectively identify the trends change point compared to the corresponding HMM.

As shown in Fig 4, we choose Anguilla, Algeria, Paraguay and Republic of Moldova for national case analysis and testing. By comparing the original RCA data, the country-product matrix

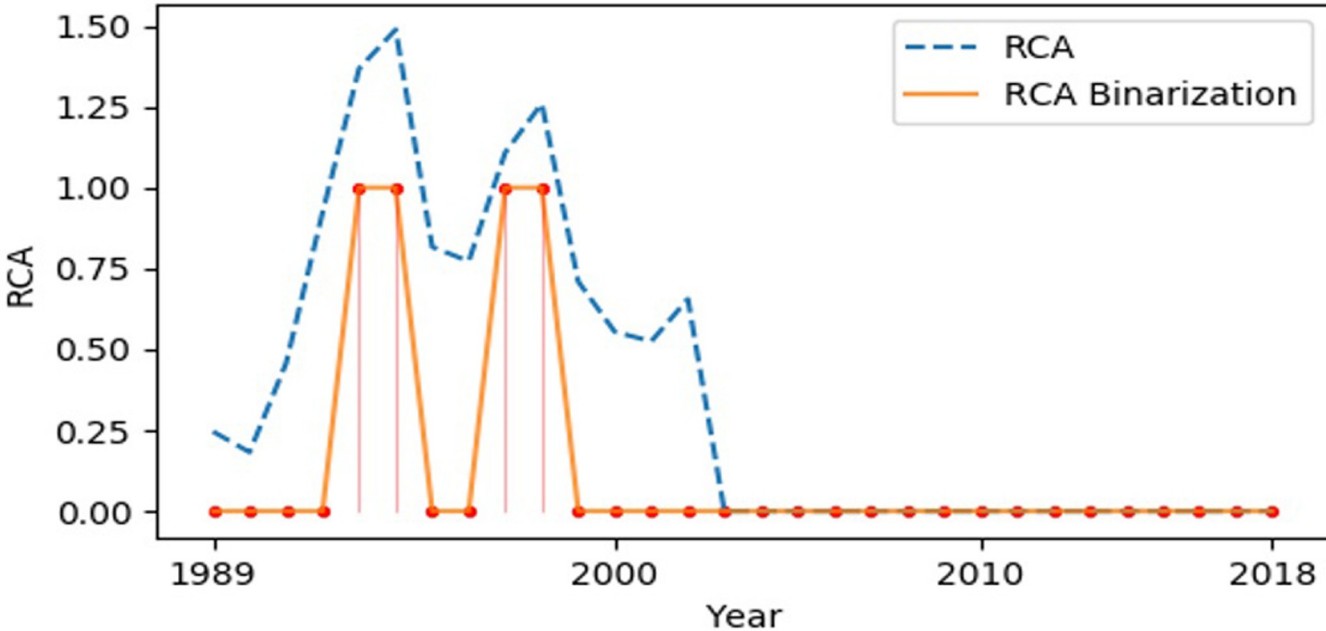

**Fig 3. RCA raw data (RCA) and RCA binarization data (RCA Binarization).** The data analysis is performed by comparing the RCA raw data (RCA) and RCA binary noise reduction data (RCA Binarization), in which the dotted line is the original RCA data and the solid line is the RCA binary noise reduction data.

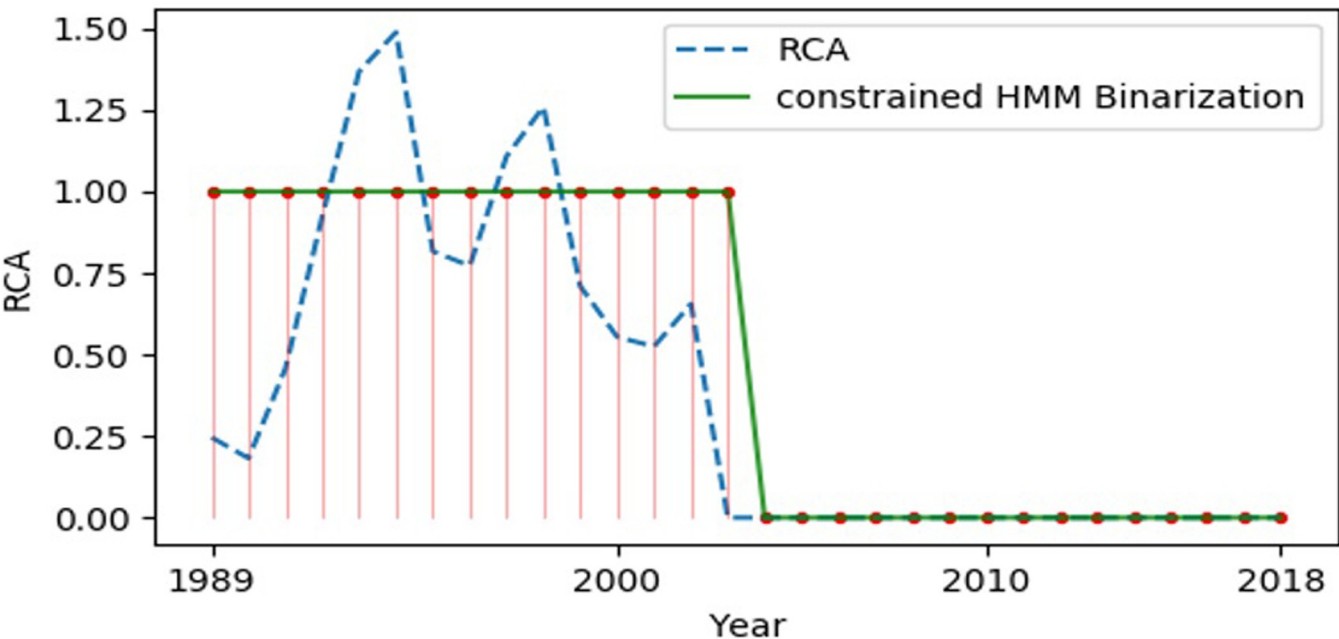

**Fig 4. RCA raw data (RCA) and SDC-HMM data (constrained HMM Binarization).** The data analysis is performed by comparing the RCA raw data (RCA) and the constrained HMM Binarization data (constrained 1-11VIM Binarization), in which the dotted line is the original RCA data and the solid line the constrained HMM Binarization data.

binarized noise reduction data, the unconstrained Hidden Markov Model noise reduction data and the SDC-HMM reduction data, a total of four kinds of results are processed for noisy data. Finally, the test for data noise reduction and change point detection are performed, which improves the predictability of economic data and the effectiveness of subsequent empirical prediction.

A great deal of country case studies demonstrate that SDC-HMM can erase abnormally high value (low value) noise in the stationary phase, abnormally low value (high value) in the rising phase and abnormally high value (low value) noise during the descending period. The data can also be significantly optimized for noise reduction, compared to country-product matrix binary noise reduction and unconstrained HMM. SDC-HMM not only makes a difference when dealing with a single type of data noise, it also realizes multiple types of data noise reduction in the same time series data. As shown in the group diagram for the Republic of Moldova in Fig 5, the SDC-HMM can effectively remove the abnormal low value noise in the rising period and the abnormal low value noise in the descending period. The use of the SDC-HMM can achieve noise reduction of two different types of noise data through the constraint condition "t = 2" and improve data quality.

Furthermore, in the country case test, we find that the SDC-HMM can retain the advantages of change point recognition. As shown in Fig 6, the Republic of Armenia and the United Kingdom of Great Britain and Northern Ireland are typical illustrations.

## Test 3: Noise estimation test

In order to further strengthen the quantitative test and analysis of the degree of noise reduction of SDC-HMM data, we conduct noise estimation tests on the country-product matrix and the SDC-HMM. The noise estimate approach was originally utilized by Tacchella et al. [21] to

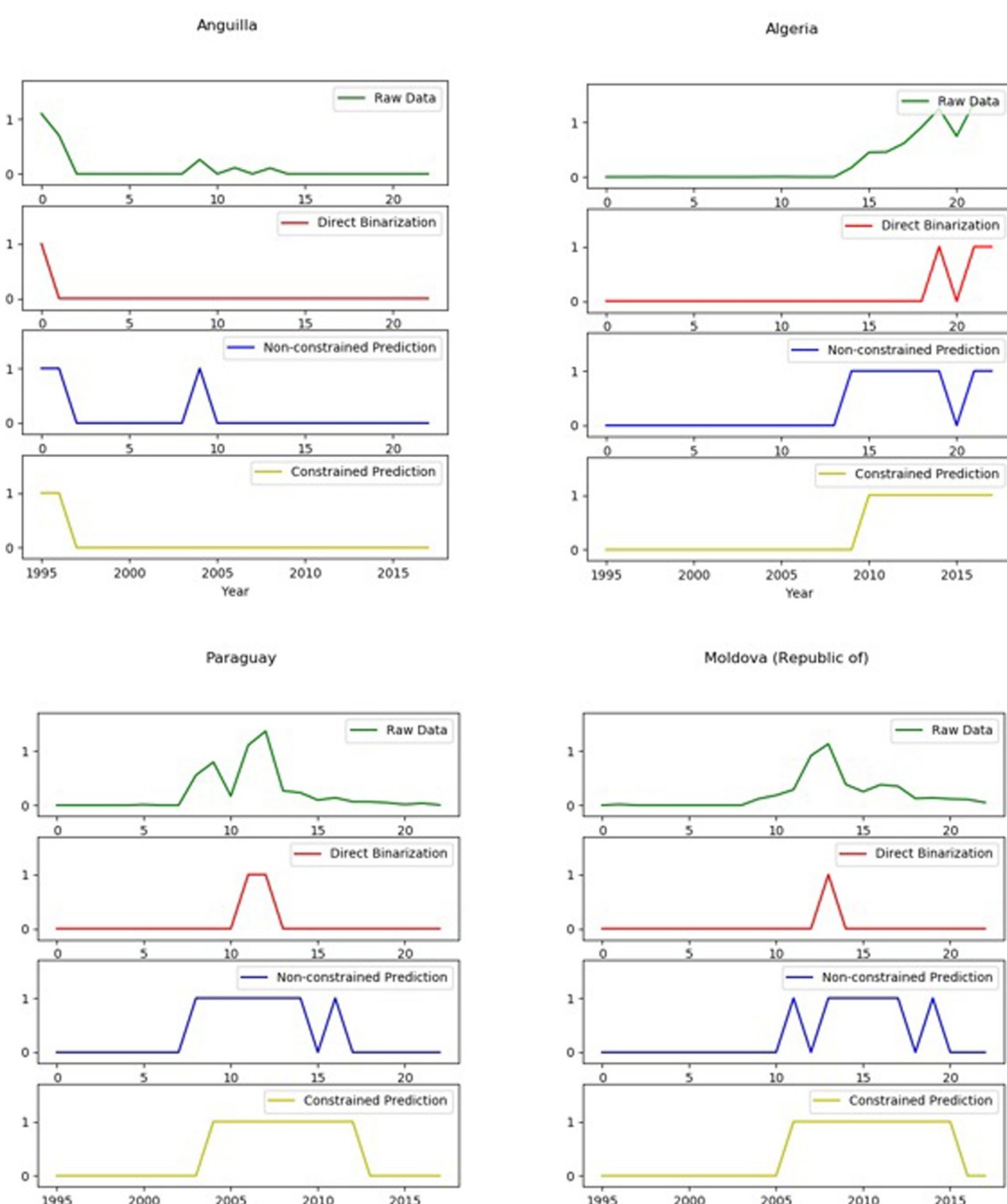

**Fig 5.** Country case test (A), including (1) original RCA data (2) country-product matrix binarized noise reduction data, (3) unconstrained Hidden Markov Model noise reduction data and (4) the Shortest Duration Constrained Hidden Markov model reduction data These show the abnormal data noise sanitation of SDC-HIAI. The horizontal axis is the time axis ("0" represents the starting year 1995, -20" represents 2015): and the vertical axis is the corresponding value of the sub-graph_ The original RCA data graph is the original value, and the rest of the sub-graphs are binary values.

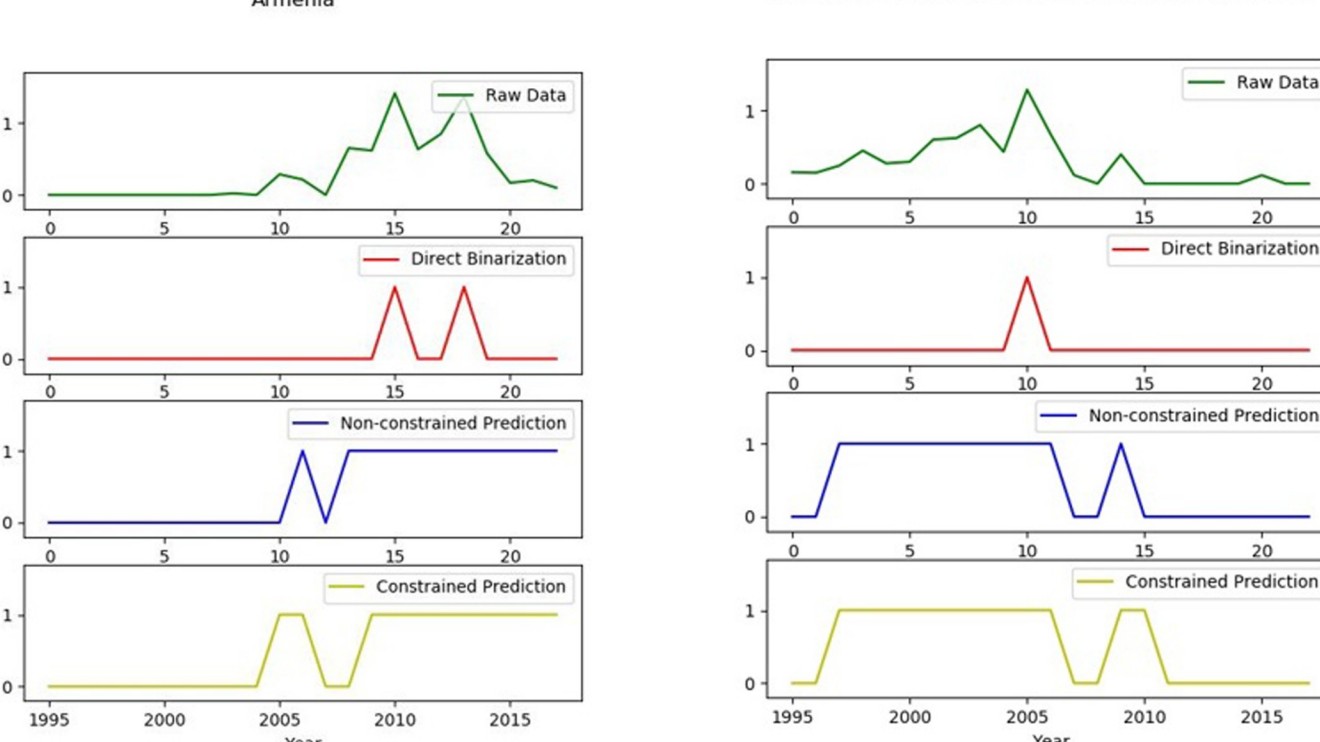

**Fig 6.** Country case test (B), including (1) original RCA data, (2) country-product matrix binarized noise reduction data, (3) unconstrained Hidden Markov Model noise reduction data and (4) the Shortest Duration Constrained Markov Model reduction data. These show the long-term change point identification of SDC-HMM.

compare the noise reduction effect of the HH index method [2] and their proposed non-linear index method (Non-Linear Metrics). The detailed steps are as follows:

They calculate the corresponding Spearman's Correlation Coefficient ($\rho_s$) from real data every year.

$$\rho_s = \frac{\sum_i (x_i - \bar{x})(y_i - \bar{y})}{\sqrt{\sum_i (x_i - \bar{x})^2 \sum_i (y_i - \bar{y})^2}} \tag{6}$$

$\rho_s$ is a non-parametric indicator used to measure the dependence between a couple of statistics. The noise estimation test results are shown in Fig 7.

In Fig 7, from 1995 to 2015, that is a period of 20 years, the average data noise percentage of the RCA binarized data obtained by noise estimation is about 42%, while the average value of data noise after SDC-HMM optimization is about 32%. In comparison, after SDC-HMM optimization, the data noise is reduced by about 25%. The SDC-HMM possesses significant advantages in noise reduction not only at the average level, but also for extreme values. After the shortest constraining of HMM, the maximum value of data noise is 33.09%, which is much lower than the minimum value of RCA binary data noise of 41.67%. The noise estimation test confirms that the SDC-HMM has a significant effect on data noise reduction.

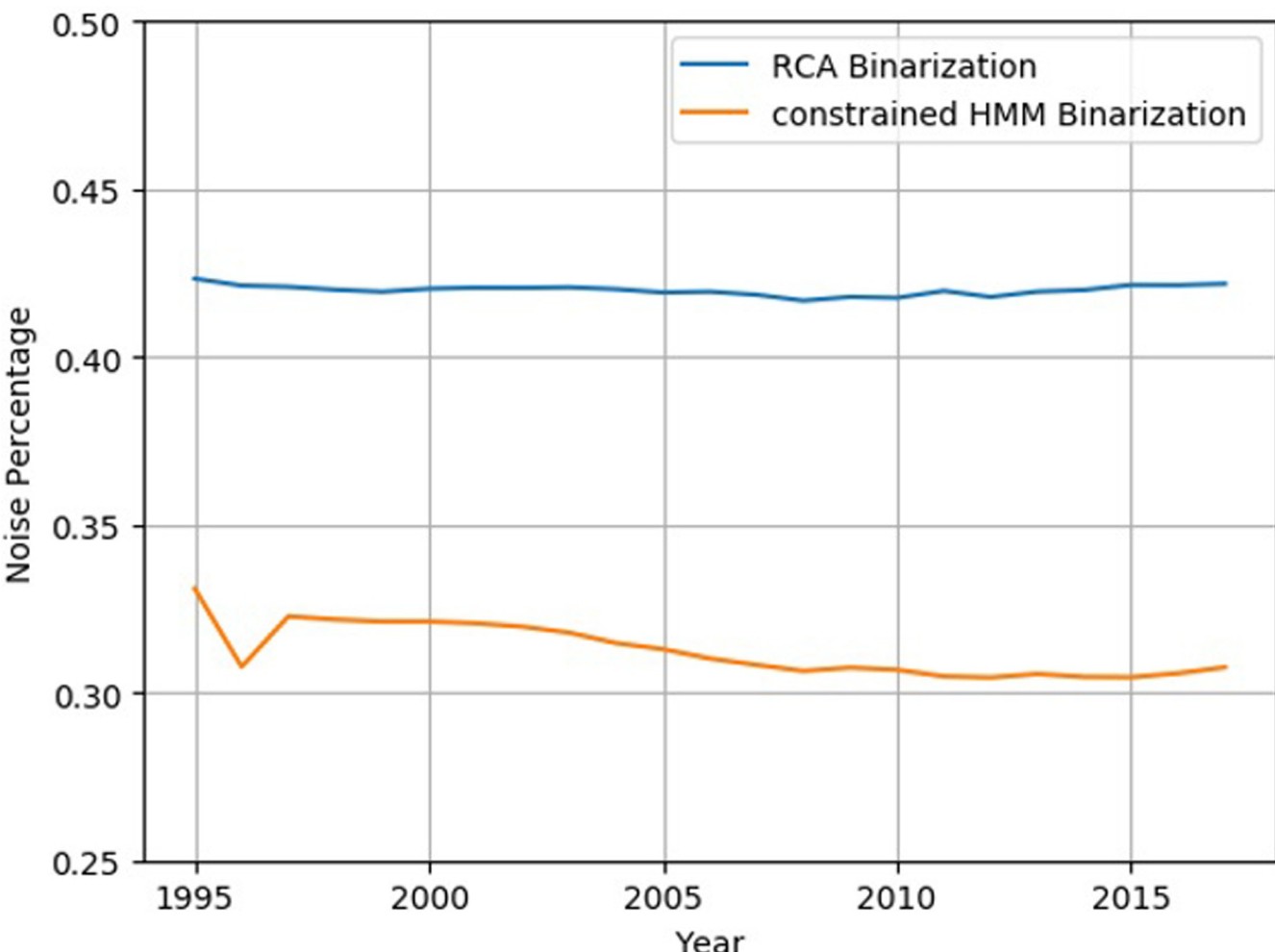

**Fig 7. Noise estimation test.** The blue line is the noise estimation of RCA binarization and the orange one is the noise estimation of SDC-HMM. The average data noise percentage of the RCA binarized data obtained by noise estimation is higher than the average value of data noise after SDC-HM optimization.

## Test 4: Fitness-GDP panel regression test

In order to study the improvement of the GDP prediction of Economic Fitness after SDC-HMM noise reduction, the panel data model is used for econometric analysis.

Based on the correlation study of GDP growth factors, the fixed effects model (FEM) constructed in this paper is as follows.

$$\ln GDP_{pc} = \alpha_{it} + \beta lnFitness_{it} + \gamma lnPop_{it} + \delta lnLF_{it} + \mu_{it} \tag{7}$$

$GDP_{pc}$, $lnFitness_{it}$, $lnPop_{it}$ and $lnLF_{it}$ represent GDP per capita, the economic fitness index, national population, and labor force over 15 years old respectively.

In Table 1, Models 1 to 3 are the mixed regression model and Models 4 to 6 are the fixed effect model after SDC-HMM noise reduction while Table 2 has the same model after HMM noise reduction. From the results of the Hausman test, we can see that because the p value of Models 5 and 6 is statistically significant in the 0.01 confidential interval, the fixed effect model should be used instead of the random effect model. Model 4 rejects the null hypothesis at a

**Table 1. Regression results of the mixed regression model and fixed effect model of economic fitness for GDP prediction after SDC-HMM noise reduction.**

| ln $GDP_{PC}$ | Model 1 | Model 2 | Model 3 | Model 4 | Model 5 | Model 6 |
|---|---|---|---|---|---|---|
| lnFitness | .3589 (0.2920) | 1.4134*** (0.0020) | .7773*** (0.0040) | 2.3270*** (0.0000) | 1.6922*** (0.0000) | 1.6997*** (0.0000) |
| lnPop | | -.5655*** (0.0000) | -3.1677*** (0.0000) | | 4.2929*** (0.0000) | -1.8903 (0.3230) |
| lnLF | | | 2.5415*** (0.0000) | | | 5.1306*** (0.0000) |
| Constant | 3.3470*** (0.0000) | 7.6490*** (0.0000) | 9.3226*** (0.0000) | 2.4500*** (0.0000) | -33.5675*** (0.0000) | -22.9120*** (0.0000) |
| $R^2$ | 0.0011 | 0.4331 | 0.5945 | 0.0103 | 0.2794 | 0.1769 |
| | | | | 0.2659 | 0.4557 | 0.5133 |
| obs | 110 | 110 | 110 | 110 | 110 | 110 |
| F statistic | 1.12 | 42.63 | 51.8 | 37.67 | 43.11 | 35.86 |
| Prob>F | 0.2923 | 0.0000 | 0.0000 | 0.0000 | 0.0000 | 0.0000 |
| F-test | - | - | - | 55.68 | 35.60 | 24.01 |
| Prob>F | - | - | - | 0.0000 | 0.0000 | 0.0000 |
| Hausman | - | - | - | 2.79 | 44.76 | 48.86 |
| Prob> $chi^2$ | - | - | - | 0.0947 | 0.0000 | 0.0000 |

Notes: * $p < 0.1$,

** $p < 0.05$,

*** $p < 0.01$; $R^2$ is adjusted for Models 1 to 3, and their significance is tested with an F-statistic; $R^2$ is the overall value (top line) and the within value (bottom line) for Models 4 to 6, and their significance is tested with the Wald Chi statistic.

significance level of 10%, which confirms the applicability of the fixed-effect model. Table 2 fits the same principle.

In order to test the robustness of the regression results, we apply the method of time-divided regression to the model, dividing the period into 1995–2007 and 2008–2017 and performing regression analysis on these two periods (Appendix A1-A4 in S1 Appendix).

**Table 2. Regression results of the mixed regression model and fixed effect model of economic fitness for GDP prediction after HMM noise reduction.**

| ln $GDP_{PC}$ | Model 1 | Model 2 | Model 3 | Model 4 | Model 5 | Model 6 |
|---|---|---|---|---|---|---|
| lnFitness | .08704 (0.813) | 1.2648*** (0.000) | .7773* (0.068) | 1.8398*** (0.000) | 1.2267*** (0.004) | 1.2399*** (0.002) |
| lnPop | | -.5507*** (0.0000) | -3.3318*** (0.0000) | | 4.8742*** (0.000) | -1.3101 (0.522) |
| lnLF | | | 2.7197*** (0.000) | | | 5.1310*** (0.000) |
| Constant | 3.4713*** (0.000) | 7.5960*** (0.000) | 9.3756*** (0.000) | 2.6780*** (0.000) | -38.2670*** (0.000) | -27.6082*** (0.000) |
| $R^2$ | 0.0005 | 0.4000 | 0.5742 | 0.0005 | 0.2958 | 0.2061 |
| | | | | 0.1246 | 0.3809 | 0.4385 |
| obs | 110 | 110 | 110 | 110 | 110 | 110 |
| F statistic | 0.06 | 35.67 | 47.64 | 14.8 | 31.68 | 26.56 |
| Prob>F | 0.823 | 0.000 | 0.000 | 0.000 | 0.000 | 0.000 |
| F-test | - | - | - | 43.17 | 32.39 | 19.57 |
| Prob>F | - | - | - | 0.000 | 0.000 | 0.000 |
| Hausman | - | - | - | 1.45 | 51.27 | 55.29 |
| Prob> $chi^2$ | - | - | - | 0.2278 | 0.000 | 0.000 |

Notes: * $p < 0.1$,

** $p < 0.05$,

*** $p < 0.01$; $R^2$ is adjusted for Models 1 to 3, and their significance is tested with an F-statistic; $R^2$ is the overall value (top line) and the within value (bottom line) for Models 4 to 6, and their significance is tested with the Wald Chi statistic.

Comparing the time-divided regression model with the initial model, the results show that the original model results are robust.

For thorough analysis of the prediction optimization of SDC-HMM, Tables 1 and 2 are compared as follows. From the mixed regression model, the results of Models 1–3 all show that SDC-HMM has better regression interpretation strength than the HMM. The significance level of the positive interpretation of the economic fitness index for GDP in Model 3 has increased to 1% of SDC-HMM, compared with 10% of HMM. From the perspective of the fixed-effect model, the standard errors and p-values of the economic fitness coefficients in all models demonstrate that economic fitness has strong explanatory power for economic growth, and it is significant at the 1% level. The regression results of SDC-HMM in Model 4 show that an increase of 1% in the economic fitness index will cause GDP to increase by 2.33%, and it is significant at the 1% level. In Model 4 of HMM, the economic fitness index is increased by 1%, which can only explain the 1.84% growth in GDP. Similarly, the SDC-HMM economic fitness indicators in Models 5 and 6 separately show that the percentage of each economic fitness indicator supporting GDP growth increased from 1.23% to 1.69% and from 1.24% to 1.70%.

### Fitness-GDP Forecast Optimization

When it comes to the forecast optimization of SDC-HMM, we propose the strictly out-of-sample forecast to verify whether the HMM with shortest duration constraint performs better than the initial HMM without constraint. We apply the Mean Absolute Error (MAE), Root Mean Squared Error (RMSE) and Mean Squared Error (MSE) between the predicted GDP and the real GDP data.

As shown in Table 3, the MAE, RMSE and MSE between the GDP predicted by the SDC-HMM and the real value of GDP in that year are reduced compared to HMM. MSE decreased by 0.89%, MAE decreased by 0.62%, and RMSE decreased by 0.44%. After the shortest duration constraint is imposed, the GDP predicted by the V-SPS (the Velocity Selective Predictability Scheme) method [6] has improved and becomes more accurate. It can be concluded that the SDC-HMM, compared with unconstrained HMM, works better in revealing the development potential of a country, thereby improving the accuracy of GDP forecasting.

### Discussion

From an economic perspective, a complex and complete industrial system and competitive export products are an inexhaustible driving force for a country's economic growth. Occupying a higher portion of the global value chains will lead to more product added value. The migration of a country's product in the chains and the acquisition or loss of the product's revealed comparative advantage are essentially the same process, and it is one that takes time to accomplish objectively. The same product of a country loses competitive advantage at one time node adjacent to another with high RCA value, and this data point violates economic development and the reality of global production. Therefore, such data should be regarded as noisy data which are meaningless for predictive analysis. The basis of economic analysis is high-quality economic data that can be utilized for prediction [22]. Furthermore, the simpler the numerical composition of economic data, the more data denoise can improve data quality. The simpler the data required for economic analysis, the more essential data quality improvement is for economic analysis. The only data required for economic fitness analysis are the "country-product matrix" data and the "country-product matrix" data which are binary data, whose numerical composition is extremely simple.

**Table 3. The results of the SDC-HMM on Fitness-GDP forecast optimization.**

|  | V-SPS(SDC-HMM) | V-SPS(HMM) | Optimization Effect |
|---|---|---|---|
| MAE | 8.0048% | 8.0548% | 0.62% |
| RMSE | 10.9453% | 10.9941% | 0.44% |
| MSE | 1.1980% | 1.2087% | 0.89% |

Notes: The Mean Absolute Error (MAE), Root Mean Squared Error (RMSE) and Mean Squared Error(MSE) on 2-years' GDP prediction.

We select economic fitness analysis and "country-product matrix" data to test the SDC-HMM for data sanitation and optimize the GDP prediction accuracy of the V-SPS method which is constructed by economic fitness. Economic Fitness is used to measure the degree of product diversification with competitiveness and product complexity in global value chains. The Fitness-Complexity Algorithm is a non-linear iteration method which can produce different results for different countries, despite them having the same initial value. Each iteration step will add more information regarding the complexity of the product, and eventually in the iteration process separate the countries with the same initial value. Further analysis means that according to the non-linear iterative recursive structure (see Eq 3), the iterative recursion algorithm of the fitness index implies the following fact: when a country's competitive products become more diversified, in the next stage the complexity of the products produced will be greater, and increased product complexity will further diversify the country's competitive products. This is a benign recursion cycle which captures the economic essence. Considering the economic logic behind the mathematical recursion progress, the calculated Fitness will further amplify noise if the original data are noisy.

We construct a constrained Hidden Markov Model with shortest duration t = 2. With the country-product matrix test, the SDC-HMM can greatly diminish the RCA raw data fluctuation noise at the "1" threshold, while retaining effective identification and confirmation of data trend change points. As confirmed by national case tests, we find that the SDC-HMM excels at removing abnormal high value (low value) noise in the stationary period, abnormal low value (high value) noise in the rising period and abnormal high value (low value) noise in the descending period, and in identifying the long-term change point of the data in low-to-high and high-to-low long-term trends, achieving obvious noise reduction. Through the noise estimation test, the SDC-HMM reduces total data noise by about 25%, which is significantly better than unconstrained HMM. In the empirical panel regression test using national economic fitness indicators to predict GDP growth, SDC-HMM is shown to improve the effectiveness of economic fitness indicators in interpreting economic growth. The Mean Squared Error (MSE) between the predicted GDP and the real GDP data decreased by 0.89%. There is two main importance of the SDC-HMM. Firstly, setting a minimum duration greater than one period is conducive to making full use of data time dimension information and effectively removing abnormal data value noise. The export data used for the Economic Fitness analysis often fluctuate drastically in a short period due to factors such as tariffs, exchange rates, and trade policies of other countries, but such changes do show the commensurable obversion of Economic Fitness. To characterize inherent economic development and predict future trends through data, the unstable state of export RCA data and Country-Product Matrix does not fit the stable GVCs. As a result, the predictive effect of a country's Economic Fitness on its economic growth diminishes too. Secondly, setting the value of d to be greater than or equal to 2 is to fully retain the information of data time dimension and identify the change of data trend in time. The data with a longer minimum duration constraint will give rise to the real variation points being missed. Furthermore, the time lag will reduce the accuracy of macroeconomic forecasting to a certain extent.

In this paper, the shortest duration constraint is defined as a constant 2 according to economic significance. When it comes to the other economic fields, this invariable duration could also be the limitation. Consequently, the more stable data requires a longer duration constraint. In addition, subsequent research could functionally set the shortest duration constraint according to the specific situation of data and the purpose of economic research. In order to improve the accuracy of economic research and prediction, the specific value or function form could be accomplished by distinguishing the degree of noise reduction through noise detection.

## Methods

The original export data used for the RCA calculation and Gross Domestic Products both come from the UN Comtrade Database [23], which was created by the United Nations Statistics Department and consists of annual trade data provided by officials from more than 200 countries and regions each year. For preparatory cleaning of the raw country's export data, this paper selects the BACI database [24] provided by the French Prospective Research and International Information Center and chooses the HS92 version of product-level international trade data. The Labor Force used for the regression test come from the World Bank databank [25]. The Populations are obtained from the Penn World Table 9.0 (PWT) produced by the University of Groningen and the University of Pennsylvania [26]. The model used for noise estimation specifically draws on the method in [21].

The computational complexity of SDC-HMM in this paper is not high. Compared with the HMM used in the EF, SDC-HMM increases the calculation amount very little. As most of the computation is applied in the parameters of HMM, the shortest duration constraint (SDC) only consumes a small amount of calculation. Compared with the calculation method of Tacchella et al. (2018), we provide a better noise reduction method without increasing the amount of calculation. It takes about 10–20 minutes to perform the SDC-HMM algorithm of the noise reduction task in this paper by using a PC.

## Supporting information

**S1 Data.**
(RAR)

**S1 Appendix.**
(DOCX)

## Author Contributions

**Conceptualization:** Xiangyu Zong.

**Data curation:** Xiangyu Zong, Qin Zhao.

**Investigation:** Pengcheng Song, Ximing Chen.

**Methodology:** Xiangyu Zong, Ximing Chen.

**Project administration:** Pengcheng Song.

**Resources:** Xiangyu Zong.

**Software:** Xiangyu Zong, Qin Zhao, Lubingzhi Guo.

**Supervision:** Pengcheng Song.

**Validation:** Ximing Chen.

**Visualization:** Ximing Chen.

**Writing – original draft:** Pengcheng Song, Ximing Chen.

**Writing – review & editing:** Pengcheng Song, Xiangyu Zong, Ximing Chen, Qin Zhao.

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
