## [Decision Letter · Decision Letter 0]

18 Nov 2020

PONE-D-20-26064

The Shortest Duration Constrained Hidden Markov Model: Data Denoise and Forecast Optimization on the Country-Product Matrix for the Fitness-Complexity Algorithm

PLOS ONE

Dear Dr. Zong,

Thank you for submitting your manuscript to PLOS ONE. After careful consideration, we feel that it has merit but does not fully meet PLOS ONE’s publication criteria as it currently stands. Therefore, we invite you to submit a revised version of the manuscript that addresses the points raised during the review process.

Please revise the paper by considering the reviewer's comments.

We look forward to receiving your revised manuscript.

Kind regards,

Jie Zhang

Academic Editor

PLOS ONE

Journal Requirements:

2.In your Data Availability statement, you have not specified where the minimal data set underlying the results described in your manuscript can be found. PLOS defines a study's minimal data set as the underlying data used to reach the conclusions drawn in the manuscript and any additional data required to replicate the reported study findings in their entirety. All PLOS journals require that the minimal data set be made fully available. For more information about our data policy, please see http://journals.plos.org/plosone/s/data-availability.

Reviewers' comments:

Reviewer's Responses to Questions

**Comments to the Author**

1. Is the manuscript technically sound, and do the data support the conclusions?

Reviewer #1: Partly

2. Has the statistical analysis been performed appropriately and rigorously? 

Reviewer #1: Yes

3. Have the authors made all data underlying the findings in their manuscript fully available?

Reviewer #1: Yes

4. Is the manuscript presented in an intelligible fashion and written in standard English?

Reviewer #1: Yes

5. Review Comments to the Author

Reviewer #1: This paper investigates the shortest duration constrained Hidden Markov Model, which is applied in data denoise and forecast optimization on the country-product matrix for the fitness-complexity algorithm. The structure of the article is not very clear and legibility is poor. My comments are as follows:\\\\

1) What is the contribution of this paper? The innovation points of the research are not clear.\\\\

2) Why $X_2$ in the Hidden Markov Model is different from the other parameters?\\\\

3) How to choose the value of $d$? Does it have specific meaning in the economic field?\\\\

4) Why can $d$ be set to improve the effect?\\\\

5) The symbols do not change before and after standardization in page 5 line 157.\\\\

6) Most of the pictures are not sharp enough.\\\\

7) With the proposed method in this paper, the computational complexity of the algorithm needs to be analyzed.

6. PLOS authors have the option to publish the peer review history of their article (what does this mean?). If published, this will include your full peer review and any attached files.

Reviewer #1: No

---

## [Author Response · Author response to Decision Letter 0]

30 Jan 2021

Thanks to your careful and valuable comments. We will explain your concerns point by point.

Reviewer #1:

Q1: What is the contribution of this paper? The innovation points of the research are not clear.\\\\

A1: The possible contributions of this paper are added in page 3, and we divide them into four different aspects. The main innovations of this paper are also added in page 3, and we divide them into two different aspects.

Q2: Why $X_2$ in the Hidden Markov Model is different from the other parameters?\\\\

A2: The $X_2$ is incorrectly capitalized, and we have corrected it in page 3 line 72.

Q3: How to choose the value of $d$? Does it have specific meaning in the economic field?\\\\

A3: The choice of this value of $d$ aims to improve the economic forecasting effect through two aspects, they are introduced in the Results. The value of $d$ has specific meaning in the economic field, the economic characteristics and implications are introduce in the Results.

Q4: Why can $d$ be set to improve the effect?\\\\

A4: The reason lies in the strong economic characteristics and implications of $d$ in economic time series data, please see in Results too.

Q5: The symbols do not change before and after standardization in page 5 line 157.\\\\

A5: The symbols after standardization are misspelled, and we have corrected them in page 5 line 157.

Q6: Most of the pictures are not sharp enough.\\\\

A6: Some pictures are modified to make them more sharp.

Q7: With the proposed method in this paper, the computational complexity of the algorithm needs to be analyzed.

A7: We briefly introduce the computational complexity of the algorithm in the Methods

---

## [Decision Letter · Decision Letter 1]

26 Apr 2021

PONE-D-20-26064R1

The Shortest Duration Constrained Hidden Markov Model: Data Denoise and Forecast Optimization on the Country-Product Matrix for the Fitness-Complexity Algorithm

PLOS ONE

Dear Dr. Zong,

Thank you for submitting your manuscript to PLOS ONE. After careful consideration, we feel that it has merit but does not fully meet PLOS ONE’s publication criteria as it currently stands. Therefore, we invite you to submit a revised version of the manuscript that addresses the points raised during the review process.

Please revise the paper by considering the reviewer's comments.

We look forward to receiving your revised manuscript.

Kind regards,

Jie Zhang

Academic Editor

PLOS ONE

Journal Requirements:

Reviewers' comments:

Reviewer's Responses to Questions

**Comments to the Author**

1. If the authors have adequately addressed your comments raised in a previous round of review and you feel that this manuscript is now acceptable for publication, you may indicate that here to bypass the “Comments to the Author” section, enter your conflict of interest statement in the “Confidential to Editor” section, and submit your "Accept" recommendation.

Reviewer #2: (No Response)

2. Is the manuscript technically sound, and do the data support the conclusions?

Reviewer #2: Yes

3. Has the statistical analysis been performed appropriately and rigorously? 

Reviewer #2: Yes

4. Have the authors made all data underlying the findings in their manuscript fully available?

Reviewer #2: (No Response)

5. Is the manuscript presented in an intelligible fashion and written in standard English?

Reviewer #2: Yes

6. Review Comments to the Author

Reviewer #2: (No Response)

7. PLOS authors have the option to publish the peer review history of their article (what does this mean?). If published, this will include your full peer review and any attached files.

Reviewer #2: No

---

## [Author Response · Author response to Decision Letter 1]

7 Jun 2021

Thanks to your careful and valuable comments. We will explain your concerns point by point.

Q1: The keywords should reflect the main research areas. The two main effects of the SDC-HMM both should be involved. Please specify them clearly.

A1: The two main effects of the SDC-HMM both have been involved. The new keywords: country-product matrices; constrained Hidden Markov Model; data denoise; forecast optimization.

Q2: The introduction part is organized with order, but in order to give readers a better picture I suggest introducing more details of your approach i.e. SDC-HMM, and stressing the importance of it.

A2: We have added a detailed description of the SDC-HMM method and explained the importance of the SDC-HMM method in two aspects. First, setting a minimum duration greater than one period is conducive to making full use of data time dimension information and effectively removing abnormal data value noise. Second, setting the value of d to be greater than or equal to 2 is to fully retain the information of data time dimension and identify the change of data trend in time.

Q3: Page 13, the authors claim that “It takes about 10-20 minutes to perform the SDC-HMM algorithm of the noise reduction task in this paper by using a PC.” The computational complexity should be further discussed, for example, how is it compared with previous literature?

A3: We have further discussed the computational complexity. As most of the computation is applied in the parameters of HMM, the shortest duration constraint (SDC) only consumes a small amount of calculation. Compared with the calculation method of Tacchella et al. (2018), we provide a better noise reduction method without increasing the amount of calculation. 

Q4: Figures need brief notes and so for Tables in order to indicate the readers enough information. Please add essential notes for each table.

A4: We have added brief but essential notes to all the figures and tables.

Q5: The discussion part should be further improved. The key findings, importance and limitations should be further discussed.

A5: The key findings, importance and limitations have been further discussed in the discussion part. we find that the SDC-HMM excels at removing abnormal high value (low value) noise in the stationary period, abnormal low value (high value) noise in the rising period and abnormal high value (low value) noise in the descending period, and in identifying the long-term change point of the data in low-to-high and high-to-low long-term trends, achieving obvious noise reduction. The limitation is for the shortest duration constraint is defined as a constant 2, which could be different in other economic fields.

Q6: Please add a full stop or comma (if the sentence is not finished) symbol behind each equation.

A6: A full stop or comma symbol has been added behind each equation, in which the sentence is not finished.

Misprints

Q1: Page 13, Footnote 1, “mmodity” should be “Commodity”.

A1: The misprint has been corrected.

Q2: Page 13, Footnote 2, “Band” should be “Bank”.

A2: The misprint has been corrected.

---

## [Decision Letter · Decision Letter 2]

15 Jun 2021

The Shortest Duration Constrained Hidden Markov Model: Data Denoise and Forecast Optimization on the Country-Product Matrix for the Fitness-Complexity Algorithm

PONE-D-20-26064R2

Dear Dr. Zong,

We’re pleased to inform you that your manuscript has been judged scientifically suitable for publication and will be formally accepted for publication once it meets all outstanding technical requirements.

Kind regards,

Jie Zhang

Academic Editor

PLOS ONE

Additional Editor Comments (optional):

Reviewers' comments:

Reviewer's Responses to Questions

**Comments to the Author**

1. If the authors have adequately addressed your comments raised in a previous round of review and you feel that this manuscript is now acceptable for publication, you may indicate that here to bypass the “Comments to the Author” section, enter your conflict of interest statement in the “Confidential to Editor” section, and submit your "Accept" recommendation.

Reviewer #2: All comments have been addressed

2. Is the manuscript technically sound, and do the data support the conclusions?

Reviewer #2: Yes

3. Has the statistical analysis been performed appropriately and rigorously? 

Reviewer #2: Yes

4. Have the authors made all data underlying the findings in their manuscript fully available?

Reviewer #2: Yes

5. Is the manuscript presented in an intelligible fashion and written in standard English?

Reviewer #2: Yes

6. Review Comments to the Author

Reviewer #2: (No Response)

7. PLOS authors have the option to publish the peer review history of their article (what does this mean?). If published, this will include your full peer review and any attached files.

Reviewer #2: No

---

## [Editor Report · Acceptance letter]

14 Jul 2021

PONE-D-20-26064R2 

The Shortest Duration Constrained Hidden Markov Model: Data Denoise and Forecast Optimization on the Country-Product Matrix for the Fitness-Complexity Algorithm 

Dear Dr. Zong:

I'm pleased to inform you that your manuscript has been deemed suitable for publication in PLOS ONE. Congratulations! Your manuscript is now with our production department. 

Kind regards, 

on behalf of

Dr. Jie Zhang 

Academic Editor

PLOS ONE